# Temocillin: Applications in Antimicrobial Stewardship as a Potential Carbapenem-Sparing Antibiotic

**DOI:** 10.3390/antibiotics11040493

**Published:** 2022-04-07

**Authors:** Tommaso Lupia, Ilaria De Benedetto, Giacomo Stroffolini, Stefano Di Bella, Simone Mornese Pinna, Verena Zerbato, Barbara Rizzello, Roberta Bosio, Nour Shbaklo, Silvia Corcione, Francesco Giuseppe De Rosa

**Affiliations:** 1Unit of Infectious Diseases, Cardinal Massaia, 14100 Asti, Italy; francescogiuseppe.derosa@unito.it; 2Department of Medical Sciences, Infectious Diseases, University of Turin, 10126 Turin, Italy; ilaria.debenedetto@edu.unito.it (I.D.B.); giacomo.stroffolini@unito.it (G.S.); simone.mornesepinna@unito.it (S.M.P.); barbara.rizzello@unito.it (B.R.); roberta.bosio@unito.it (R.B.); nour.shbaklo@edu.unito.it (N.S.); silvia.corcione@unito.it (S.C.); 3Department of Medical, Surgical and Health Sciences, University of Trieste, 34127 Trieste, Italy; stefano932@gmail.com; 4Infectious Diseases Unit, Trieste University Hospital (ASUGI), 34125 Trieste, Italy; verena.zerbato@gmail.com; 5School of Medicine, Tufts University, Boston, MA 02111, USA

**Keywords:** temocillin, antimicrobial stewardship, sparing strategy

## Abstract

Temocillin is an old antibiotic, but given its particular characteristics, it may be a suitable alternative to carbapenems for treating infections due to ESBL-producing *Enterobacterales* and uncomplicated UTI due to KPC-producers. In this narrative review, the main research question was to summarize current evidence on temocillin and its uses in infectious diseases. A search was run on PubMed using the terms (‘Temocillin’ [Mesh]) AND (‘Infection’ [Mesh]). Current knowledge regarding temocillin in urinary tract infection, blood-stream infections, pneumonia, intra-abdominal infections, central nervous system infections, skin and soft tissues infections, surgical sites infections and osteoarticular Infections were summarized. Temocillin retain a favourable profile on microbiota and risk of *Clostridioides difficile* infections and could be an option for treating outpatients. Temocillin may be a valuable tool to treat susceptible pathogens and for which a carbapenem could be spared. Other advantages in temocillin use are that it is well-tolerated; it is associated with a low rate of *C. difficile* infections; it is active against ESBL, AmpC, and KPC-producing *Enterobacterales*; and it can be used in the OPAT clinical setting.

## 1. Introduction

In recent decades, the rise of multidrug-resistant (MDR) bacteria has become one of the greatest challenges in global health. Extended-spectrum β-lactamase (ESBL)-producing *Enterobacterales* and carbapenem-resistant *Enterobacterales* (CRE) are considered the main threats worldwide [1,2,3]. Carbapenems have become the first empiric choice for treating severe infections in settings with a high prevalence of ESBL and AmpC-producing bacteria in order not to delay effective antibiotic treatment. Unfortunately, the increasing consumption of carbapenems has led to the rising selection and dissemination of CRE [4,5]. Thus, there is increasing interest in the pipeline of new antibiotics coupled with the reassessment of older agents from the perspective of a carbapenem-sparing strategy [6]. Temocillin is a semisynthetic 6-a-methoxy derivative of ticarcillin (Figure 1, molecular formula C_16_H_18_N_2_O_7_S_2_) a penicillin developed, used intravenously at usual dose of 2 gm q12h (2 gm q8 h in critically ill patients), and commercialised in the United Kingdom and Belgium in the 1980s. 

As a result of its structure, temocillin presents unusual stability against ESBL β-lactamases and AmpC-derepressed mutants of Enterobacteriaceae, maintaining bactericidal activity toward these bacteria even if slower compared with susceptible strains [7,8,9]. It also retains activity against *Haemophilus influenzae*, *Moraxella catarrhalis*, *Neisseria* spp., and *Burkholderia cepacia*. The affinity to penicillin-binding protein (PBP) 1, PBP2, and PBP3 is reduced, but temocillin binds tightly to PBP5 and PB6, partially explaining the lack of activity against Gram-positive cocci, anaerobes, and nonfermenting Gram-negatives, such as *Pseudomonas aeruginosa* and *Acinetobacter baumannii*, [7,8,10,11]. Some strains of *P. aeruginosa* identified in patients with cystic fibrosis harboured mutations (mexA, mexB) restoring emocillin susceptibility in 15% of the strains [12]. The protein binding is high (80%), and the half-life after intravenous (IV) infusion is nearly five hours. The main elimination route is renal via glomerular filtration, and only a small fraction is eliminated via tubular excretion, suggesting that dosing should be corrected in renal impairment [13]. The urinary recovery after 24 h ranged from 72 to 82%. Animal studies found that fT > MIC correlated with the maximum efficacy of the drug [14,15]. Temocillin is highly dialysable, with a fraction eliminated by dialysis of approximately 55% [13]. Vandecasteele et al. proposed a three-times-weekly schedule (2 g every 48 h), during which the free serum concentration remained above the MIC as high as 50–90%, even for MICs of 16 mg/L [13,16]. Temocillin was found stable at 37 °C for 24 h, suggesting that prolonged or continued infusion dosing could be suitable, particularly for critically ill patients [17]. 

Temocillin is quite well tolerated. No neurological adverse effects are described [18,19]. Carbapenems, piperacillin/tazobactam, amoxicillin/clavulanate, cefepime, ceftazidime, and ciprofloxacin are chemically incompatible with temocillin. Vancomycin, clindamycin, and clarithromycin are physically incompatible with temocillin [17]. There are no data about the safety of temocillin during human pregnancy. 

Temocillin is bactericidal, and its activity is only slightly affected by inoculum size [13]. Moreover, unlike most cephalosporins, temocillin does not select AmpC-derepressed variants [20]. High MICs are reported for temocillin against CRE, particularly those producing OXA-48 and/or metalloenzymes (e.g., IMP, NDM, and VIM producers), and less than 10% of isolates retain susceptibility, except for urinary *Klebsiella pneumoniae* carbapenemase (KPC)-producing *Enterobacterales*, in which susceptibility was demonstrated in almost 85% of strains [21,22]. EUCAST, recently, defined susceptibility breakpoints for temocillin: MIC ≤ 16 mg/L for uncomplicated UTI caused by *E. coli*, *Klebsiella* spp. (except *K. aerogenes*), and *P. mirabilis*; MIC ≤ 8 mg/L for other infections and species; and validated on disc diffusion a zone diameter < 17 mm for resistance [23]. The BSAC (British Society for Antimicrobial Chemotherapy) has set MIC ≤ 8 mg/L as a breakpoint for systemic infections and MIC ≤ 32 mg/L for UTI. [24]. 

Temocillin is an old antibiotic, but given its particular characteristics, it may be a suitable alternative to carbapenems treating infections due to ESBL-producing *Enterobacterales* and uncomplicated UTI due to KPC-producers. This review aims to provide clinical data on potentially valuable applications of temocillin in clinical practice.

## 2. Results

### 2.1. Temocillin in Urinary Tract Infections

The urinary excretion of unchanged temocillin is near 80%, mainly with a minimal rate of tubular secretion after intravenous administration and 80–92% after intramuscular (IM) administration [25,26]. Temocillin achieves a concentration of 400–600 mg/L in urine, making it an attractive choice for UTI [27]. In a murine model of UTI due to ESBL-producing Escherichia coli, an initial bacterial killing followed by regrowth was seen at a concentration equal to MIC, but at a concentration that exceeds the MIC 4-fold, the bactericidal activity was sustained, and an almost maximal bactericidal effect was observed for values of fT/MIC over 80% [14]. In this study, the authors suggest that the standard twice-a-day regimen (200 mg/kg in 2 h for 2 g q12h) could be useful for treating pathogens with a breakpoint of 16 mg/L, but a three-times-a-day administration (200 mg/kg q4 and q6h) is suggested for an MIC of 32 mg/L [14].

Older studies have demonstrated excellent bactericidal activity against uropathogens, such as Enterobacterales, with a low rate of resistance, but reduced or no activity against non-fermenters, including *P. aeruginosa* [7,27,28,29,30,31,32,33,34,35] (Table 1).

More recently, more than 90% of Enterobacterales harbouring AmpC and ESBL remained susceptible to temocillin using the urinary breakpoint (MICs ≤ 32 mg/L) [11,36,37]. In vitro, at the urinary breakpoint, 2280 strains of Enterobacterales with a high rate of carbapenemase-producers were tested for susceptibility to temocillin. Overall, 77.1% of isolates were susceptible to temocillin, including 93% KPC-producing strains, while Enterobacterales harbouring other carbapenemases, such as OXA-48 or metallo-β-lactamases, were resistant in 91% of cases [38]. 

A low rate of synergistic activity with aminoglycosides was seen against Gram-negative species, but additive effects were demonstrated in 22% of strains [8]. Kitzis et al. demonstrated that the susceptibility to temocillin was only slightly affected by the presence of CTX-1, TEM, and β-lactamases [39]. In another study, temocillin was demonstrated to be stable against ESBL-producing *E. coli* with porin mutation, including strains harbouring chromosomal AmpC β -lactamases [40]. Two clinical trials are currently ongoing, NCT03543436 [41] and NCT04478721 [42], with the aim of comparing temocillin and carbapenems in cUTI due to Gram-negative bacteria resistant to third-generation cephalosporins.

Moreover, compared with carboxypenicillins and ureidopenicillins, temocillin was found to be highly stable against enterobacterial inducible β-lactamases [10]. Studies carried out in the past decades have demonstrated high rates of cure in uncomplicated UTI but high rates of relapse and clinical failure in complicated urinary tract infections (cUTI), including pyelonephritis, after 7–10 days IV treatments [43,44]. On the contrary, Shulze et al. reported a complete response in seven patients with pyelonephritis treated for 7–10 days with both dosages of 500 or 1000 mg twice a day of temocillin [45]. In a single Intensive Care Unit (ICU), Offenstadt et al. reported 11 patients with UTI, including six with sepsis, being treated with temocillin. The most common bacteria isolated were *Enterobacterales*, mainly *E. coli* [46]. The clinical cure was achieved in six patients, two patients had clinical failure, and three patients died. Interestingly, in one of them, a second urine culture revealed the presence of *P. aeruginosa* [46]. The efficacy of temocillin in children with cUTI was retrospectively evaluated in 22 children with a mean age of 5.8 years. Twenty-one out of 22 children had acute pyelonephritis, mainly caused by *E. coli* [47]. A bacterial cure was observed for all temocillin-susceptible strains. A multicentre retrospective study in the United Kingdom (UK) evaluated the efficacy of temocillin in 92 patients with different infections, including 42 UTI patients. The overall rate of ESBL/AmpC producing strains was 58%. Clinical and microbiological cures were achieved at 90% and 87%, respectively. Interestingly, the clinical efficacy was strongly improved when 2 g twice daily (instead of 1 g twice daily) was used [48].

### 2.2. Temocillin in Bloodstream Infections (BSI)

Temocillin is approved in Europe for the treatment of bacteraemia, UTI, and lower respiratory tract infections at a posology of 2 g twice daily; nonetheless, it is available for intravenous use in the UK, Belgium, Germany, and France only [13,27,49,50,51,52]. A recent study [35] tested 400 isolates, including 260 ESBL- or AmpC-producing isolates and 40 KPC-producing isolates, and found that 61.8% of the isolates were susceptible to temocillin using the BSAC breakpoint for systemic infections (≤8 mg/L). Among the KPC-producing isolates, even though more than one-third were susceptible to temocillin according to UTI breakpoint, all of them were considered resistant when systemic infection breakpoint was applied. Another study investigated 42 BSI in 92 patients with infections due to *Enterobacterales* treated with temocillin, where 53 of the overall isolates were ESBL or derepressed AmpC producers, and the cure rate was 84% [45]. To date, no comparative study between temocillin and carbapenems or other antibiotics in infections caused by ESBL- or AmpC-producing Enterobacteriaceae has been published. The optimal dosage in BSI is unknown, even though higher cure rates were reported in patients treated with temocillin at the dosage of 2 g twice daily versus <2 g twice daily, with a more pronounced difference in the ESBL or derepressed AmpC subset [45]. A resistance rate of 69% to temocillin in BSI has been described in the multidrug-resistant ST131-O25b clone of *E. coli* in a multicentric study in the UK and Republic of Ireland; among these isolates, the most frequently detected ESBL was CTX-M-15 (87%) [53].

Alexandre et al. [54] recently investigated the use of temocillin in France, reporting rates of clinical failure in UTI and non-UTI. They reported significant differences in clinical failure rates in sepsis compared with severe sepsis (6%) or septic shock (25%) treated with temocillin. The authors did not observe differences between 2 g q12h and 2 g q8h doses in clinical failure rates or between *E. coli*, *Klebsiella* spp., *Proteus* mirabilis, and other Enterobacterales [54].

### 2.3. Temocillin in Pneumonia

Clinical data on temocillin used in community-acquired or hospital-acquired pneumonia are scarce, as well as data about epithelial lining fluid (ELF)/plasma penetration ratios [55,56]. Despite the lack of activity against Gram-positive microorganisms and Gram-negative non-fermenters such as *A. baumannii, Burkholderia cepacia* and *P. aeruginosa*, synergistic combination regimens, including temocillin, have been proposed within in vitro studies with ampicillin, flucloxacillin, and ticarcillin to enhance anti-Pseudomonal or anti-Staphylococcal activity [57,58].

In a retrospective audit, Habayeb et al. [59] reviewed 192 episodes of hospital-acquired pneumonia treated with piperacillin/tazobactam versus amoxicillin plus temocillin, and no difference in the clinical success rate was observed between the two groups. Nonetheless, a significant inferior rate of diarrhoea and *Clostridioides difficile* infection were reported in patients treated with amoxicillin plus temocillin. 

Recently, Layios et al. have described 32 patients who were treated for VAP with intermittent infusion or continuous infusion of 6g of temocillin daily for in vitro susceptible pathogens [60]. However, continuous infusion showed superior PK/PD indexes, despite that fall short of current recommendations for systemic infections, save for moderate renal impairment [60].

Continuous infusion of a dose of temocillin of 4 g/day was tested in a randomised control trial among ICU patients with nosocomial pneumonia. In continuous infusion, the drug remained stable for 24 h and compatible with flucloxacillin and aminoglycosides co-administration. Even though stable free serum concentrations above the breakpoint of 16 mg/L were yielded, the authors suggest that lowering the breakpoint to 8 mg/L may be warranted because of individual variations in this population [17]. Currently, no ongoing clinical trial has been registered to date to investigate the use of temocillin in carbapenem-sparing strategies in patients with hospital-acquired pneumonia. 

### 2.4. Temocillin in Abdominal Infections

The penetration of temocillin into bile and peritoneal fluid is high, and this evidence provides the basis for its use in intra-abdominal infections.

Pfeiffer et al. first described the therapeutic efficacy of temocillin, in 30 critically ill patients, including adults suffering from peritonitis and intra-abdominal abscesses [53]. In this study, the patients were treated with 1 g temocillin administered intravenously twice daily. The isolated pathogens comprised mainly *E. coli* and *Proteus*, but *Enterococci*, *Pseudomonas* spp., *Klebsiella*
*pneumoniae*, *Citrobacter* spp., *Bacteroides* spp., strepto, and *Peptococcus* spp. were also implicated. Temocillin was reported to be effective in 21 out of the 22 patients with peritonitis, as well as in six out of eight patients with long-lasting infections due to temocillin-sensitive pathogens. No adverse reactions to temocillin were observed [61].

Temocillin penetrated rapidly, and during the first hour post administration, the peritoneal level was 48% of the serum level. The mean peritoneal level of temocillin over the study period (3.5 h) was 49.1 mg/L. It was concluded that 1 g of temocillin twice daily would achieve sufficiently high intraperitoneal levels to inhibit susceptible pathogens [62].

Wittke et al. studied temocillin at a dosage of 2 g twice daily in 25 biliary surgery patients in whom potential septic complications were a concern [63]. Clinical efficacy was assessed as ‘very good’ in 23 patients. In one patient, there was a disorder of wound healing, and in another, staphylococcal bronchial pneumonia developed postoperatively [63]. Temocillin was tolerated very well, and no side effects were observed. Twelve hours after the administration of 2 g of temocillin intravenously to surgical patients, the mean serum concentration was 22.44 (+/− 10.26) mg/L. The median half-life was 3.86 (+/− 1.84) h. Mean concentrations of 12.44 and 38.59 mg/L were measured up to the 12th hour in the wound secretions and peritoneal secretions, respectively. In skin, fat, fascia, muscle, and gallbladder wall, temocillin concentrations greater than the inhibitory concentrations of most Gram-negative bacteria were demonstrated after one and two hours [63].

More recently, Berleur et al. described the activity of combination therapy between fosfomycin and temocillin in vitro and in vivo in a murine peritonitis model against *E. coli* strains producing KPC-3 or OXA-48-type carbapenemases. This combination prevented the emergence of fosfomycin resistance and proved to be more bactericidal than fosfomycin alone [64].

Similarly, Alexandre et al. described that in a murine infection model with bacteraemia from intra-abdominal origin, temocillin retained significant activity in peritoneal fluid, blood, and spleen and prevented death in mice by effectively working against KPC-producing *E. coli* with temocillin MICs ≤ 16 mg/L [54].

### 2.5. Temocillin in Central Nervous System (CNS) Infections

A limited number of papers have been published exploring the use of temocillin in the setting of central nervous system (CNS) infections [65,66,67,68]. The cerebrospinal fluid (CSF)/blood penetration of temocillin is deemed to range between 8 and 15% of the plasma concentration, resulting higher in patient with meningitis [65,66]. This data come from a single study by Bruckner et al., which assessed almost 40 years ago the diffusion of temocillin in CSF in four neurosurgical patients with external ventricular drains and four patients with meningitis [66]. In the study, temocillin was given 2 g twice daily, and the analysis was conducted with high-performance liquid chromatography. No CSF temocillin accumulation was observed. [19,66]. The authors conclude that temocillin CSF concentrations in these subjects seemed to be inadequate for the treatment of Gram-positive bacterial meningitis and only partially valuable for the treatment of Gram-negative bacillary meningitis, but these conclusions are hard to make inference from, as no data are available regarding MIC for identified pathogens [66]. In a simulated study of a rabbit model, an infusion system was applied to the assessment of therapeutic efficacy in an experimental infection model for meningitis. The results indicated the potential for this system in experimental studies but did not provide useful information regarding the CNS penetration of the molecule or its possible place in therapy [67]. A single case report explored the possibility of treating complicated epidural abscesses caused by an ESBL-producing *K. pneumoniae* with temocillin [68]. No case of CNS adverse effects that can be normally attributed to other beta-lactams has been registered for temocillin [19].

### 2.6. Temocillin in Skin, Soft Tissues, Surgical Sites, and Osteoarticular Infections

Few data are available on skin, soft tissues, surgical sites, and osteoarticular infections. One case study reports the use of temocillin in treating peripheral phlebitis in a *K. pneumoniae* infection complicating a psoas abscess by *S. aureus* [69]. Similarly, few studies have explored the use of temocillin in osteoarticular settings. Two cases have reported the use of temocillin in knee synovitis caused by *Pantoea agglomerans* and cervical osteomyelitis caused by *Burkholderia cepacia* [70,71]. Another model suggested the use of the molecule in antibiotic-loaded bone cement, as temocillin retained its antimicrobial activity after elution from the bone cement [72]. 

### 2.7. Temocillin in Venereal and Sexual Transmitted Diseases

No data exist for temocillin with regard to treponemal infections. Still, by a microbiological point of view, the molecule should be active. According to an in vitro study, temocillin might be an effective agent against *Chlamydia trachomatis* [73]. That would be of added value in view of a possible monotherapy targeting the two most common venereal diseases that are frequently concomitant, with little influence by the inoculum size and with a single dose and low side effects [74]. In fact, temocillin is active against *Neisseria gonorrhoeae* and resistant to beta-lactamase produced by this pathogen. Specifically, it is active against both penicillinase-producing and non-penicillinase-producing strains [75]. Moreover, in a recent danish study, it has not been possible to provide ceftriaxone- resistant *N. gonorrhoeae* to test temocillin, as the prevalence of ceftriaxone-resistant *N. gonorrhoeae* is low in the area [76]. Overall, the activity rate was high, and resistance to other molecules may guide temocillin susceptibility [77]. Due to the long half-life of temocillin, a single I.M. dose may be a suitable option for treating STIs, possibly targeting both *C. trachomatis* and *N. gonorrhoeae*.

### 2.8. Outpatient Antibiotic Treatment (OPAT)

Growing interest has been registered for the use of beta-lactams in the outpatient clinical setting. The reason for that relies on the possibility of avoiding unnecessary hospital acquired infections, reducing cross-infections in high-risk patients, to reduce care-associated costs of hospitalisation, while still keeping the advantages of the high activity of the selected molecules and the complete longue-course therapies traditionally carried on entirely in hospital. To date, attempts have mainly been made with ceftolozane/tazobactam (C/T), a novel cephalosporin/β-lactamase inhibitor, that showed good stability at room temperature and was found to be safe, effective, and convenient in the *P. aeruginosa* OPAT [66,67]. Temocillin is stable at 25°C and should be properly diluted in sterile water [78,79]. In a prospective randomized, controlled pharmacokinetic study, 32 patients with infections caused by *Enterobacterales* received temocillin 2 g every 8 h or 6 g in continuous infusion. Mean, median, and range of percentages of the dosing interval during which the free drug concentration remained >16 mg/L were 76.4, 98, and 18.7–98.9 in patients treated three times daily and 98.9, 89.7, and 36.4–99.9 in patients with continuous infusion, respectively [80]. Temocillin has shown also to be stable also in elastomeric pumps and possibly particularly interesting in the cystic fibrosis patients treated with OPAT [78]. This novel treatment regimen could be an option for patients to avoid hospital admission or discharge to complete therapy as an outpatient, especially when targeting identified susceptible pathogens, corroborating a strategic role for this molecule in the antimicrobial stewardship perspective [79]. 

### 2.9. Impact on Microbiome

Beta-lactams are amongst the most impacting treatments when it comes to dismicro-biosis, and their over and misuse has contributed to frightening and growing data on anti-microbial resistances. Temocillin is not only less prone to be associated with *C. difficile* infection, a feared complication largely attributable to microbiome alteration, as evident from real life and animal models data [81], unlike clindamycin or cefoxitin. However, it has shown to result in less disturbances of the intestinal microbiota with respect to other commonly used beta-lactams when treating specifically UTI [82]. Moreover, the effect on the colonisation resistance was measured in a mouse model [83]. The evaluated parameters indicated a selective decontamination effect and that the drug can be used safely without an increased risk of overgrowth by resistant bacteria causing superinfections. The same study subsequently challenged the issue in 10 healthy volunteers: in none of the volunteers did the colonisation resistance appear to be affected, and selective decontamination was recorded in seven. Accordingly, in healthy volunteers, a seven-day course of temocillin did not impact on the total count of strict anaerobes and resulted in a dramatic decrease in the faecal counts of *Enterobacteriaceae*—without selecting for temocillin-resistant strains—and a concomitant overgrowth of enterococci and yeasts [84,85]. Of note, the microbiome impact of temocillin has not been reported during its clinical use. 

## 3. Discussion

The aim of this research was to summarise the current evidence on the use of temocillin in different clinical settings and to apply that knowledge in view of a carbapenem-sparing strategy in current times. We described the main PK/PD characteristics of this compound and highlighted a variety of settings in which temocillin may prove its suitability best. 

Certainly, in view of its high urinary concentration and good real-life outcomes, temocillin may be considered for targeted therapy in UTI (Figure 2). 

Exceptions are infections in which non-fermenters are confirmed. Few data are available for its use in combination therapy, especially in cUTI, in view of a potential synergistic effect that has not yet been observed. No conclusions can be drawn upon the data we retrieved from our search pertaining to empirical therapy. Assuredly, an extended infusion and higher dosages may add efficacy when using this molecule, and this is even truer for infections other than UTI, specifically concerning pneumonia, for which more data about the PK characteristics of temocillin could lead to a more tailored use [19,82]. In fact, we found little data on ELF concentration, and for pneumonia studies are incredibly scarce. This is surprising in view of the fact that beta-lactams are used extensively for pneumonia. More data are needed to assess the potential of temocillin in lung medicine, with special attention paid to it because currently, complicated CAP, HAP, ventilated HAP, and VAP are still fields in which carbapenems are much debated. The same can be said for what pertains to ICU patients, in which extended infusion and real-time TDM may lead to better PK/PD parameters, especially when concomitant CVVH is applied [82]. 

Moreover, temocillin has proven to have good penetration into peritoneal membranes and the biliary tract. The majority of data from abdominal infections point to good PK/PD characteristics and to a strong bactericidal activity. Data regarding anaerobic bacteria are poor, and no conclusions can be added based on the literature we read. Wrapping up, temocillin may be a potential therapeutic option in treating intra-abdominal infections. Moreover, the potential for combination therapy with fosfomycin exists, as suggested by preliminary data [64].

Little is known about the use of temocillin outside UTI, pneumonia, abdominal infections, and BSI settings. Treating other infections with temocillin seems to be less appealing in view of the relatively low prevalence of susceptible pathogens; furthermore, carbapenems are less commonly prescribed outside the above debated indications, with the exception of CNS infections, and when this is the case, the reason is usually the presence of a non-fermenter pathogen (e.g., *P. aeruginosa*), for which temocillin would not add any advantage. 

Overall, it may be stated that temocillin has very low side effects, good tolerability, and a low rate of *C. difficile* infection, characteristics that are highly desirable when using a beta-lactam. 

To date, there is a lack of data to effectively demonstrate the role of temocillin as a suitable alternative to carbapenems in treating infections due to ESBL-producing Enterobacterales and especially in uncomplicated UTIs due to KPC-producers [86]. High MICs are reported for temocillin against CRE, and less than 10% of isolates retain susceptibility, but urinary KPC-producing *Enterobacterales* are important exceptions: in this subset, susceptibility was confirmed in almost 85% of strains [21,22], and more in vitro data corroborate this possibility [35]. Currently, the two ongoing clinical trials, NCT03543436 [38] and NCT04478721 [39], were developed with the aim of comparing temocillin and carbapenems in cUTI due to Gram-negative bacteria resistant to third-generation cephalosporins. Obtaining data from these studies and putting temocillin in direct comparison with carbapenems may solve a long-debated issue.

In addition, certain molecules that have proven highly beneficial in real-life settings and are valuable tools in stewardship and carbapenem-deprescribing programmes have been temporarily excluded by manufacturers (i.e., C/T), and further molecules are not readily available in countries where they are needed more [87]. Recently, IDSA also provided a major update on therapeutical indications for difficult-to-treat infections, specifically pneumonia, and a prescribing shift must be operated from broad non-targeted carbapenem strategies to a more innovative approach in prescribing and de-prescribing [88]. Temocillin may be included and repurposed in this thoughtful approach, probably scaling up diagnostic tools, implementing combination strategies, and ultimately providing alternatives in the path of stewardship programmes. 

## 4. Materials and Methods

The current narrative review followed five steps: identifying the research question, selecting search methods for identifying relevant studies, study selection, charting and summarizing data, and reporting the results. The main research question was to summarize current evidence on temocillin and its uses in infectious diseases. A search was run on PubMed using the terms (‘Temocillin’ [Mesh]) AND (‘Infection’ [Mesh]) in English. Results were limited to 1 December 1980 and 1 February 2022. A list of 87 papers was generated from the initial search. Then reviewers studied titles and abstracts. Finally, quality assessment of full-text studies was performed by two independent reviewers (IDB and TL). Researchers reviewed the summary of all articles sought and ultimately used data from 70 full articles to compile this review paper. Researchers assessed for inclusion all titles and abstracts without language limitations in English. We included papers that described evidence on temocillin and its clinical uses in infectious diseases. We excluded papers that had no methods described, duplicated other studies previously included, or were not strictly related to temocillin (Figure 3).

## 5. Conclusions

In conclusion, temocillin may be a valuable antibiotic to treat pathogens that are susceptible and for which a carbapenem could be spared, especially in UTIs, pneumonias, and IAIs. Other advantages are its safety; its low rate of *C. difficile* infections; its activity against ESBL, AmpC, and KPC producing *Enterobacterales*; and the OPAT clinical setting. On the other hand, clinical breakpoints and optimal dosages are still a matter of debate, and more clinical studies are needed to reignite a revival. 

## Figures and Tables

**Figure 1 antibiotics-11-00493-f001:**
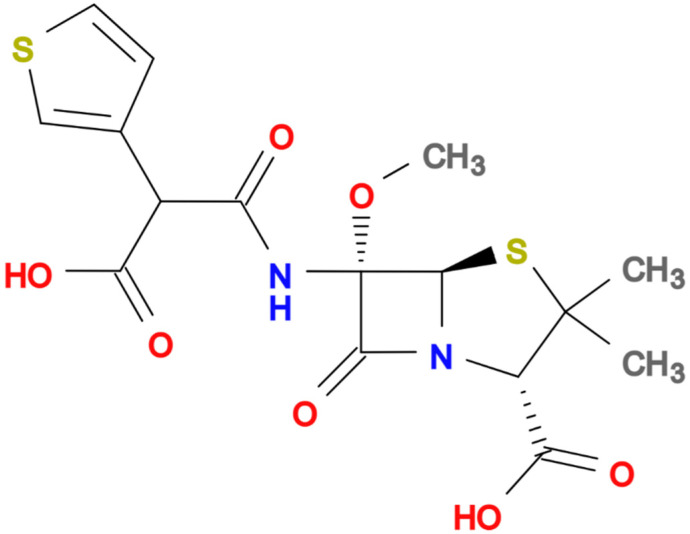
Chemical structure of Temocillin.

**Figure 2 antibiotics-11-00493-f002:**
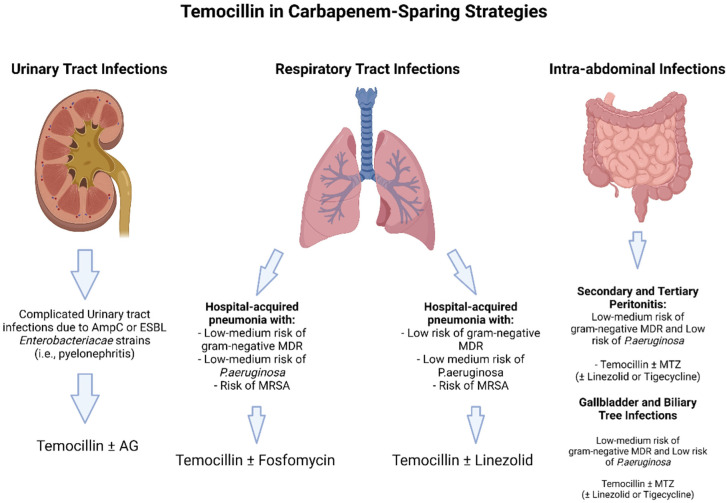
Theoretical carbapenem sparing regimens including Temocillin. Abbreviations: ESBL: extended spectrum Beta-lactamases; AG: aminoglycosides: MDR: multi-drug resistant; MRSA: methicillin resistant *Staphylococcus aureus*; MTZ: metronidazole.

**Figure 3 antibiotics-11-00493-f003:**
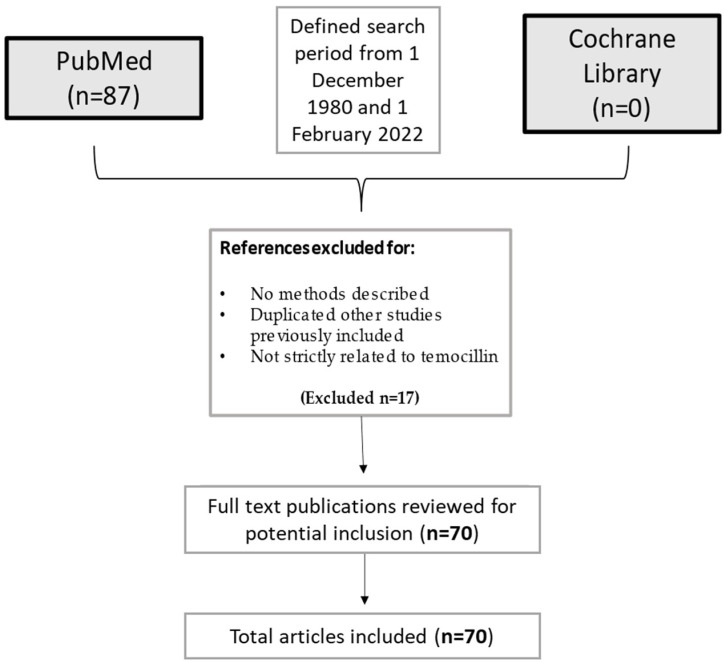
Flow-chart of the studies considered in the narrative review.

**Table 1 antibiotics-11-00493-t001:** Clinical studies and Clinical experiences with Temocillin in Infectious Diseases.

Author, Year, and Reference	Study Design	Number of Patients	Antibiotic and Dosing	Source of Infection	Isolates	Clinical Outcomes
Kosmidis J, 1985	Interventional Study	33	Temocillin, 500 mg q24h (IM) or 1 g q24h (IV), for 7 to 10 days	UTI and cUTI	*E. coli* (24), *P. mirabilis* (7), and *E. cloacae* (2)	Temocillin IM (Clinical cure 83% in UTI, Not effective in cUTI); IV (Clinical Cure 100% in UTI, 70% in cUTI)
Asbach HW et al., 1985	Interventional Study	29	Temocillin, 500 mg q12h (IV), for 5 to 7 days	UTI and cUTI	*E. coli* (20), *Proteus* spp. (9), *Klebsiella* spp. (4), *E. faecalis* (2), *S. epidermidis* (1), and *P. stuartii* (1)	Clinical and microbiological cure 93%
Schulze B et al., 1985	Open Clinical Study	20	Temocillin, 500 mg q12h (IV) for 7 to 10 days or 1 g q12h (IV), for 7 to 15 days	UTI, cUTI, LRTIs, and BSI	*E. coli* (14), *M. catarrhalis* (3), *P. vulgaris* (2), *K. oxytoca* (1), *H. influenzae* (1), *H. haemolyticus* (1), and *E. aerogenes* (1)	Clinical cure 100% in both groups
Lindsay G et al., 1985	Interventional Study	32	Temocillin, 1 g q12h (IV or IM) for 7 to 14 days	UTI, cUTI, and LRTIs	*E. coli* (6), *Klebsiella* spp. (9), *Enterobacter* spp. (4), *Proteus mirabilis* (2), *C. freundii* (1), and *H.alveii* (1)	Clinical cure 78%
Pfeiffer et al., 1985	Retrospective Study	30	Temocillin, 1 g q12h (IV)	IAIs, SSTIs, and LRTIs	*E. coli* (17), *Proteus* spp. (5), *Enterococci* (3), *Pseudomonas* spp. (3), *K. pneumoniae* (3), *Citrobacter* spp. (2), *Bacteroides* spp. (2), *Streptococci* (1), and *Peptococci* (1)	Clinical cure 77%
Legge et al., 1985	Interventional Study	13	Temocillin, 500 mg q12h (IV) or 1 g q12h (IV) or 2 g q12h (IV) for 7 to 10 days	LRTIs	*E. coli*, *Klebsiella* spp., *Acinetobacter* species, *P. mirabilis*, *H. influenzae,* and *H. haemoglobinophilus*	Clinical cure 84.6%
Gray et al., 1985	Interventional Study	16	Temocillina, 2–3 g day for 5–10 days	LRTIs	*H. influenzae* (8) and *S. pneumoniae* (5)	Clinical cure 81.25%
Saylam et al., 2002	Case Report	1	NA	Vertebral Osteomyelitis, Pyomiositis, and CRBSI	*K. pneumoniae*, *S. aureus*	Complete clinical cure
Lekkas et al. 2005	Interventional Study	23	Temocillin, 2–6 g day, 14 (range 1–40)	CF	*B. cepacia*	Clinical Improvement 56.25%
Duerinckx, 2008	Case Report	1	Temocillin, 1 g q12h (IV) for 6 days	Synovitis	*Pantoea agglomerans*	Complete clinical cure
Barton et al., 2008	Case Report	1	Temocillin, 2 g q12h (IV) for 12 weeks	Epidural abscess	ESBL *K. pneumoniae*	Complete clinical cure
Gupta et al., 2009	Retrospective Study	6	Temocillin, 1 g q24h (IV), from 4 days to 24 months	UTI, cUTI, LRTIs, IAIs, and BSI	*Klebsiella* spp. (4), *E. coli* (1), and *E. aerogenes* (1)	Clinical cure 66%
Balakrishnan et al., 2011	Retrospective Study	92	Temocillin 1 g q12h (IV) or 2 g q12h (IV)	UTI, cUTI, LRTIs, IAIs, and BSI	ESBL and/or dAmpC Enterobacterales (53)	Clinical Cure 86%; Microbiological Cure 84%
Rodriguez et al., 2013	Case Report	1	NA	Osteomyelitis	*B. cepacia*	Complete clinical cure
Habayeb et al., 2015	Interventional Study	188	Temocillin, 2 g q12h (IV) for 5–7 days vs. PTZ 4.5 g q8h for 5–7 days	LRTIs	NA	Clinical cure 82%
Laterre et al., 2015	Randomized controlled Trial	32	Temocillin, 2 g q8h (IV) or 6 g (continous infusion) or CVVH	IAIs and LRTIs	*E. coli* (13), *Klebsiella* spp. (7) or *Enterobacter* spp. (5)	Clinical cure 79% (8 h), 93% (continuous infusion), and 75% (CVVH)
Alexandre et al., 2021	Retrospective Study	153	Temocillin 2 g q8h (IV) or 2 g q12h (IV)	UTI, cUTI, LRTIs, IAIs, bone infections, and BSI	Enterobacterales (67.5% ESBL-producers)	Early Clinical Failure (UTI: 4.9%; non-UTI: 13.8%), Late Clinical Failure (UTI: 26.7%; non-UTI: 33.3%)
Heard et al., 2021	Retrospective Study	205	Temocillin, 2 g q12h (IV)	UTI, cUTI, LRTIs, IAIs, bone infections, and BSI	*E. coli* (81.1% ESBL), Non-*E. coli* Enterobacterales (41.4% AmpC, 41.4% ESBL, 2.9% KPC)	Treatment Failure at 30 days: 20.5%
Delory et al., 2021	Multicenter retrospective case-control study	144	Temocillin, 2 g q12h (IV) vs. Carbapenems (Ertapenem, Meropenem or Imipenem) (IV)	UTI and cUTI	ESBL Enterobacterales [*K. pneumoniae* (59), *E. coli* (57), and *Enterobacter* spp. (24)]	Clinical cure 94% (Temocillin Groups) and 99% (Carbapenem Comparators)
Edlund et al., 2022	Randomised, multicentre, superiority, open-label, phase 4 trial	152	Temocillin 2 g q8h (IV) or Cefotaxime 1–2 g q8h, for 7 to 10 days	UTI e cUTI	*Citrobacter* spp. (5), *Enterobacter* spp. (2), *Proteus* spp. (5), *Pseudomonas* spp. (2), *S. aureus* (1), *Aerococcus* spp. (5), *E. faecalis* (5)	Clinical Cure 98%

Abbreviations: IV: intravenous; IM: intramuscular; spp: species; UTI: urinary tract infections: LRTI: lower respiratory tract infection; SSTI: skin and soft tissue infections; IAI: intra-abdominal infections; BSI: bloodstream infections; CRBSI: catheter-related BSI.

## Data Availability

Not applicable.

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
