# Peer review of "Temocillin: Applications in Antimicrobial Stewardship as a Potential Carbapenem-Sparing Antibiotic"

_antibiotics, 2022, doi:10.3390/antibiotics11040493_

Round 1

Reviewer 1 Report

The article provide clinical data on potentially valuable applications of temocillin in clinical practice. Overall, authors should complete missing information in results section,  major changes in the context/connection of results and discussion section along with improved presentation of results.
Major comments:

  • Abstract not in style of journal.
  • Please add figure of chemical structure of Temocillin for example:
  •  
  • Molecular Formula C16H18N2O7S2
  • Authors must add tables summerize applications of Temocillin in Infectious Diseases from pervoius studies.
  • Numbered titles under results section.
  • Impact on microbiome section need to increase.
  • If you can add section about Temocillin in venereal or sexuall infection.
  • Very important part is absence to demonstrate relation between Temocillin as suitable alternative to carbapenems in treating infections due to ESBL-producing Enterobacterales and uncomplicated UTI due to KPC-producers.

Author Response

Dear Editor,

Thanks for the opportunity to revise our manuscript. Please find the response to reviewers’ comments on our manuscript entitled “Temocillin: Applications in antimicrobial stewardship as a po-tential carbapenem sparing antibioticto be considered for publication in Antibiotics

Reviewer 1# comments

  1. The article provide clinical data on potentially valuable applications of temocillin in clinical practice. Overall, authors should complete missing information in results section,  major changes in the context/connection of results and discussion section along with improved presentation of results.

Dear reviewer thank you for these comments that improve notably our manuscript. We have made the changes requested and we added missing information accordingly to your suggestions.

Major comments:

  1. Abstract not in style of journal

Dear reviewer thank you for these comments that improve notably our manuscript. We have made the changes requested in the Abstract section. Please find highlighted changes in the text.

  1. Please add figure of chemical structure of Temocillin for example:

Dear reviewer, thank you for this comment. We have added a new Figure 1.

  1. Authors must add tables summerize applications of Temocillin in Infectious Diseases from pervoius studies.

Dear reviewer, thank you for this comment. We have added a new Table that summarizes available Clinical Studies regarding Temocillin in Infectious Diseases, accordingly to your suggestions

  1. Numbered titles under results section.

Dear reviewer thank you for these comments that improve notably our manuscript. We have made the changes requested

  • Impact on microbiome section need to increase.

Dear reviewer, thank you for this comment. We have improved the microbiome section accordingly to your suggestions

  • If you can add section about Temocillin in venereal or sexuall infection.

Dear reviewer, thank you for this comment. We have added a Venereal Diseases section accordingly to your suggestions

  • Very important part is absence to demonstrate relation between Temocillin as suitable alternative to carbapenems in treating infections due to ESBL-producing Enterobacterales and uncomplicated UTI due to KPC-producers.

Dear reviewer, thank you for this comment. We have improved the discussion section accordingly to your suggestions

Reviewer 2 Report

The article is unique and on a relevant topic. however, need work to have it better presented. my comments below

Comment 1:

The title is inaccurate   “Temocillin in Current Antimicrobial Stewardship Programmes” because it is not currently used in clinical practice that much and is not mentioned in any IDSA guidelines I have read. I don’t like how the title is structured.

I recommend changing it to something like” Temocillin: Applications in antimicrobial stewardship as a potential carbapenem sparing antibiotic” or “Temocillin: revisiting an old antibiotic as carbapenem sparing strategy” etc

Comment 2:

Introduction: In the first paragraph you mentioned carbapenems as the standard of care in this situation. However, you did not mention the role of new agents like cefiderocol as carbapenem sparing agents in these notorious organisms. Please add information about cefiderocol.

Comment 3:

In the introduction, you did not mention if this is oral? Or Parenteral? Usual dose?

Comment 4:

Number each section: 1. Introduction, 2. Results, etc

Comment 5:

Results:

The results is difficult to follow since it is all text. I highly recommend that you create a Table and summarize brief information about the included studies if the reader wants a quick snapshot of your article. For example

Study disease state region and hospital name patient population regimen used, and comparator 
1        
2        

Then, put the details in the manuscript

Comment 6:

Add sub-headings; for example, “Temocillin in Urinary Tract Infections” please number and italize

2.1 Temocillin in Urinary Tract Infections

Comment 7:

Line 235: “The authors conclude temocillin CSF con” please change to “The authors concluded that temocillin CSF con” in reporting results from other studies, please used the past tense throughout the manuscript

Comment 8:

Discussion:

This part below is unnecessary and out of the scope of the review. What is the association between increasing carbapenem use and the COVID-19 pandemic! even if it does increase, because of malpractice, and it was not evidence-based medicine.  

This whole part should be deleted. And exchange it with simple, straightforward information about the increasing rate of antibiotic resistance and the need for new antibiotics. As simple as that.

“In view of recent data, carbapenem therapies have once again undergone a rise in 328 many clinical settings, and their use has also been favored by the concomitant COVID-329 19 pandemic [72], in which less tailored approaches and less commonly specialist-guided 330 antibiotics have largely been used. During a time in which MDR bacteria prevalence is 331 increasing, available drugs need to be reassessed, and their place in therapy needs to un-332 dergo a critical evaluation process.”

Comment 9:

You did not add limitations to the discussion section, please do so.

Comment 10:

Conclusion: line 361: please change “temocillin may be a valuable tool” to “temocillin may be a valuable antibiotic”, also this applies throughout the manuscript since it is an antibiotic, not a tool

Author Response

Turin, March 27th 2022

Dear Editor,

Thanks for the opportunity to revise our manuscript. Please find the response to reviewers’ comments on our manuscript entitled “Temocillin: Applications in antimicrobial stewardship as a po-tential carbapenem sparing antibioticto be considered for publication in Antibiotics

Reviewer 2# comments

  1. The article is unique and on a relevant topic. however, need work to have it better presented. my comments below

Dear reviewer thank you for these comments that improve notably our manuscript. We have made the changes requested and we added missing information accordingly to your suggestions.

Comment 1:

  1. The title is inaccurate “Temocillin in Current Antimicrobial Stewardship Programmes” because it is not currently used in clinical practice that much and is not mentioned in any IDSA guidelines I have read. I don’t like how the title is structured. I recommend changing it to something like” Temocillin: Applications in antimicrobial stewardship as a potential carbapenem sparing antibiotic” or “Temocillin: revisiting an old antibiotic as carbapenem sparing strategy” etc

Dear reviewer thank you for these comments. We have made changes requested in the text accordingly to your suggestions. We changed the title to “Temocillin: Applications in antimicrobial stewardship as a potential carbapenem sparing antibiotic”.

  • Introduction: In the first paragraph you mentioned carbapenems as the standard of care in this situation. However, you did not mention the role of new agents like cefiderocol as carbapenem sparing agents in these notorious organisms. Please add information about cefiderocol.

Dear reviewer, thank you for this comment. This manuscript is focused on Temocillin and a wider description of new agents is out of the scope of the narrative review. Despite that this could be the basis of a new project or manuscript regarding Temocillin and carbapenem strategies.

2 In the introduction, you did not mention if this is oral? Or Parenteral? Usual dose?

Dear reviewer thank you for these comments that improve notably our manuscript. We have made the changes requested and we added missing information accordingly to your suggestions.

3 Number each section: 1. Introduction, 2. Results, etc

Dear reviewer thank you for these comments that improve notably our manuscript. We have made the changes requested

4 The results is difficult to follow since it is all text. I highly recommend that you create a Table and summarize brief information about the included studies if the reader wants a quick snapshot of your article. For example:

Dear reviewer thank you for these comments. We have added Table 1 in the text accordingly to your suggestions

Add sub-headings; for example, “Temocillin in Urinary Tract Infections” please number and italize. 2.1 Temocillin in Urinary Tract Infections

Dear reviewer thank you for these comments that improve notably our manuscript. We have made the changes requested

Comment 7:

Line 235: “The authors conclude temocillin CSF con” please change to “The authors concluded that temocillin CSF con” in reporting results from other studies, please used the past tense throughout the manuscript

Dear reviewer thank you for these comments. We have made changes requested  in the text accordingly to your suggestions

Comment 8:

Discussion:

This part below is unnecessary and out of the scope of the review. What is the association between increasing carbapenem use and the COVID-19 pandemic! even if it does increase, because of malpractice, and it was not evidence-based medicine.   

This whole part should be deleted. And exchange it with simple, straightforward information about the increasing rate of antibiotic resistance and the need for new antibiotics. As simple as that.

“In view of recent data, carbapenem therapies have once again undergone a rise in 328 many clinical settings, and their use has also been favored by the concomitant COVID-329 19 pandemic [72], in which less tailored approaches and less commonly specialist-guided 330 antibiotics have largely been used. During a time in which MDR bacteria prevalence is 331 increasing, available drugs need to be reassessed, and their place in therapy needs to un-332 dergo a critical evaluation process.”

 Dear reviewer thank you for these comments. We have made changes requested  in the text accordingly to your suggestions

Comment 9:

You did not add limitations to the discussion section, please do so.

Dear reviewer thank you for these comments. We have made changes requested  in the text accordingly to your suggestions

Comment 10:

Conclusion: line 361: please change “temocillin may be a valuable tool” to “temocillin may be a valuable antibiotic”, also this applies throughout the manuscript since it is an antibiotic, not a tool

Dear reviewer thank you for these comments. We have made changes requested  in the text accordingly to your suggestions

Reviewer 2# comments

  1. The article is unique and on a relevant topic. however, need work to have it better presented. my comments below

Dear reviewer thank you for these comments that improve notably our manuscript. We have made changes requested and we added missing informations accordingly to your suggestions.

Comment 1:

  1. The title is inaccurate “Temocillin in Current Antimicrobial Stewardship Programmes” because it is not currently used in clinical practice that much and is not mentioned in any IDSA guidelines I have read. I don’t like how the title is structured. I recommend changing it to something like” Temocillin: Applications in antimicrobial stewardship as a potential carbapenem sparing antibiotic” or “Temocillin: revisiting an old antibiotic as carbapenem sparing strategy” etc

Dear reviewer thank you for these comments. We have made changes requested  in the text accordingly to your suggestions. We changed the title into “Temocillin: Applications in antimicrobial stewardship as a potential carbapenem sparing antibiotic”.

  • Introduction: In the first paragraph you mentioned carbapenems as the standard of care in this situation. However, you did not mention the role of new agents like cefiderocol as carbapenem sparing agents in these notorious organisms. Please add information about cefiderocol.

Dear reviewer, thank you for this comment. This manuscript is focused on Temocillin and a wider description of new agents is out of the scope of the narrative review. Despite that this could be the basis of a new project or manuscript regarding Temocillin and carbapenem-strategies.

2 In the introduction, you did not mention if this is oral? Or Parenteral? Usual dose?

Dear reviewer thank you for these comments that improve notably our manuscript. We have made changes requested and we added missing informations accordingly to your suggestions.

3 Number each section: 1. Introduction, 2. Results, etc

Dear reviewer thank you for these comments that improve notably our manuscript. We have made changes requested

4 The results is difficult to follow since it is all text. I highly recommend that you create a Table and summarize brief information about the included studies if the reader wants a quick snapshot of your article. For example:

Dear reviewer thank you for these comments. We have added Table 1 in the text accordingly to your suggestions

Add sub-headings; for example, “Temocillin in Urinary Tract Infections” please number and italize. 2.1 Temocillin in Urinary Tract Infections

Dear reviewer thank you for these comments that improve notably our manuscript. We have made changes requested

Comment 7:

Line 235: “The authors conclude temocillin CSF con” please change to “The authors concluded that temocillin CSF con” in reporting results from other studies, please used the past tense throughout the manuscript

Dear reviewer thank you for these comments. We have made changes requested  in the text accordingly to your suggestions

Comment 8:

Discussion:

This part below is unnecessary and out of the scope of the review. What is the association between increasing carbapenem use and the COVID-19 pandemic! even if it does increase, because of malpractice, and it was not evidence-based medicine.   

This whole part should be deleted. And exchange it with simple, straightforward information about the increasing rate of antibiotic resistance and the need for new antibiotics. As simple as that.

“In view of recent data, carbapenem therapies have once again undergone a rise in 328 many clinical settings, and their use has also been favored by the concomitant COVID-329 19 pandemic [72], in which less tailored approaches and less commonly specialist-guided 330 antibiotics have largely been used. During a time in which MDR bacteria prevalence is 331 increasing, available drugs need to be reassessed, and their place in therapy needs to un-332 dergo a critical evaluation process.”

 Dear reviewer thank you for these comments. We have made changes requested  in the text accordingly to your suggestions

Comment 9:

You did not add limitations to the discussion section, please do so.

Dear reviewer thank you for these comments. We have made changes requested  in the text accordingly to your suggestions

Comment 10:

Conclusion: line 361: please change “temocillin may be a valuable tool” to “temocillin may be a valuable antibiotic”, also this applies throughout the manuscript since it is an antibiotic, not a tool

Dear reviewer thank you for these comments. We have made changes requested  in the text accordingly to your suggestions

Round 2

Reviewer 1 Report

Thank you for improvment of article.

Only 2 minor thing:

  • in Line 48, chemical, start by Chemical.
  • Table 1, You need to add in column 1, number of refrance used at References Section.

Reviewer 2 Report

Only 1 minor thing, in Line 318, the discussion sections "3. Dicsussion" became part of the paragraph and does not represent a section. 

make sure to start new line, bolded